# Impact of Dose Escalation on the Efficacy of Salvage Radiotherapy for Recurrent Prostate Cancer—A Risk-Adjusted, Matched-Pair Analysis

**DOI:** 10.3390/cancers14051320

**Published:** 2022-03-04

**Authors:** Dirk Böhmer, Alessandra Siegmann, Sophia Scharl, Christian Ruf, Thomas Wiegel, Manuel Krafcsik, Reinhard Thamm

**Affiliations:** 1Department of Radiation Oncology, Charité University Medicine, Campus Benjamin Franklin, 12203 Berlin, Germany; alessandra.siegmann@charite.de; 2Department of Radiation Oncology, University Hospital Ulm, 89081 Ulm, Germany; sophiascharl@yahoo.de (S.S.); thomas.wiegel@uniklinik-ulm.de (T.W.); manuel.krafcsik@uniklinik-ulm.de (M.K.); reinhard.thamm@uniklinik-ulm.de (R.T.); 3Department of Urology, Bundeswehrkrankenhaus Ulm, 89081 Ulm, Germany; bwkrhsulmurologie@bundeswehr.org

**Keywords:** prostate cancer, radical prostatectomy, salvage radiotherapy, dose-escalation, matched-pair analysis

## Abstract

**Simple Summary:**

This study evaluated 554 patients who underwent radical surgery for prostate cancer and later presented with persisting or rising PSA levels which required salvage radiotherapy. Our results showed that increasing the radiation dose during radiotherapy could reduce the risk of a second PSA relapse. These findings suggested that patients with failed prostatectomies might benefit from dose-escalated salvage radiotherapy to improve tumor control and postpone secondary treatments, such as hormonal or chemotherapy.

**Abstract:**

Previous randomized trials have not provided conclusive evidence about dose escalations and associated toxicities for salvage radiotherapy (SRT) in prostate cancer. Here, we retrospectively analyzed whether dose escalations influenced progression-free survival in 554 patients that received salvage radiotherapy for relapses or persistently elevated prostate cancer antigen (PSA) after a radical prostatectomy. Patients received SRT between 1997 and 2017 at two University Hospitals in Germany. We compared patient groups that received radiation doses <7000 cGy (*n* = 225) or ≥7000 cGy (*n* = 329) to analyze the influence of radiation dose on progression-free survival. In a second matched-pair analysis of 216 pairs, we evaluated prognostic factors (pT2 vs. pT3–4, Gleason score [GS] ≤ 7 vs. GS ≥ 8, R0 vs. R1, and pre-SRT PSA <0.5 vs. ≥0.5 ng/mL). After a median follow-up of 6.8 (4.2–9.2) years, we found that escalated doses significantly improved progression-free survival (*p* = 0.0042). A multivariate analysis indicated that an escalated dose, lower tumor stages (pT2 vs. pT3/4), and lower GSs (≤7 vs. 8–10) were associated with improved progression-free survival. There was no significant effect on overall survival. Our data suggested that escalating the radiation dose to ≥7000 cGy for SRT after a prostatectomy significantly improved progression-free survival. Longer follow-ups are needed for a comprehensive recommendation.

## 1. Introduction

Among patients that undergo a radical prostatectomy (RP) for pT3 prostate cancer, persistent or rising prostate-specific antigen (PSA) levels occur in approximately 50% of those without and 70% of those with positive surgical resection margins [1,2,3,4]. Recent data have revealed that persistent PSA after an RP could significantly impact metastasis-free survival, overall survival (OS), and cancer-specific survival [3]. In non-randomized trials, the value of Prostate-Specific-Membrane-Antigen Positron-Emission-Tomography (PSMA-PET) imaging was evaluated in patients with PSA relapses [5]. Those results led to the current European Association of Urology (EAU) guideline recommendation that a PSMA-PET-CT should be offered to men with persistent PSA after RP and to men with biochemical recurrences [6].

International guidelines recommend salvage radiotherapy (SRT) for patients with persistent PSA values above 0.1 ng/mL, rising PSA values in subsequent PSA tests, or any PSA value above 0.1 ng/mL [7,8]. SRT should preferably commence before the PSA level reaches 0.5 ng/mL to ensure a high probability of achieving undetectable PSA values [9]. However, currently, we cannot rule out the possibility that there might be other indicators for SRT.

Despite new study results on SRT after RP for patients with persistent or rising PSA levels, four substantial questions remain controversial and are currently unresolved: the role of hormonal therapy, additional irradiation of pelvic lymphatics, the overall treatment dose for SRT, and the optimal fractionation scheme. Two randomized trials evaluated the addition of androgen deprivation therapy (ADT) to SRT, but the results were inconclusive [10,11]. Additionally, to date, there is no clear evidence to support the optimal treatment strategy for a PSA relapse, because there is no distinction between local, regional, distant, or both local and distant disease. Recently, a systematic review evaluated many factors to facilitate discriminations among different relapse sites [12]. Indeed, distinguishing local from distant disease is particularly important in making individual treatment decisions. Moreover, studies have shown that a high Gleason score (8–10) and a short PSA doubling time (<12 months) could significantly impact survival. Those findings may assist radiation oncologists in treatment decisions [2,12,13,14].

The role of dose escalation in SRT is currently under investigation. From primary radiotherapy, we know well that a radiation dose escalation improves oncological outcome measures, but at the expense of increased late toxicity. The Radiation Therapy Oncology Group 0126 trial confirmed these results, and furthermore, they demonstrated in a separate analysis that using appropriate intensity-modulated radiotherapy (IMRT) plans could reduce gastrointestinal or genitourinary late toxicities [15,16]. However, the oncological outcome results of two dose-escalation trials showed no significant difference between treatment arms for the primary endpoint [17,18].

In the present retrospective study, we aimed to evaluate whether dose-escalated SRT could provide an improved oncological outcome compared to lower-dose SRT. Additionally, we aimed to identify factors that could influence the oncological results.

## 2. Materials and Methods

Between 1997 and 2017, 554 patients from two university hospitals in Germany received SRT for biochemical failure after RP. We defined biochemical failure after surgery as persistent PSA levels >0.1 ng/mL or intermittent PSA elevations after undetectable PSA that rise above 0.1 ng/mL The median SRT dose was 7020 cGy, with an interquartile range (IQR) of 6660–7200 cGy. The clinical target volume comprised the prostatic fossa. When the tumor was pT3b or pT4 stage, the seminal vesicle bed was included in the SRT. Pelvic lymph nodes were not irradiated. More than half of all patients received modern radiotherapy techniques, namely IMRT. Patients were excluded when they had received a hormonal treatment before or during SRT or when they had lymph node involvement. Table 1 shows the baseline characteristics of all 554 patients.

The Cox proportional hazards regression model was used in multivariate analyses to determine and evaluate factors that could influence biochemical PFS (progression free survival). The significant risk factors were used for propensity matching [19]. The impact of dose-escalation was analyzed with the Kaplan-Meier method and univariable Cox regression [19,20].

To analyze the influence of risk factors on the outcome, we applied an adapted propensity-matching procedure with the following risk-factor groups: pT2 vs. pT3–4, GS ≤ 7 vs GS ≥ 8, R0 vs R1, and pre-SRT PSA < 0.5 vs. ≥ 0.5 ng/mL. This procedure identified 216 matched pairs.

Overall survival was defined from study initiation to death from any cause. PFS was defined as death, local or distant recurrence, initiation of any secondary anti-tumor treatment (e.g., ADT), or biochemical relapse (defined as PSA rising to more than 0.2 ng/mL above the post-SRT nadir).

## 3. Results

The median time from RP to the start of SRT was 23 months (range: 1.7–176 months). The pre-SRT PSA levels ranged from 0.04–8.87 ng/mL, with a median of 0.28 ng/mL.

After a median follow-up of 6.8 years (IQR: 4.2–9.2 years), the five-year PFS rates were 52% for patients that received <7000 cGy and 65% for patients that that received ≥7000 cGy (Figure 1). This difference was statistically significant (*p* = 0.0042).

In addition, the multivariable Cox regression analysis showed that a lower pT-stage, a lower Gleason score, a positive surgical resection status (R1), and a lower pre-SRT PSA level were significantly associated with an improved outcome (Table 2).

Figure 2a shows the PFS of propensity-matched patients. The dose-escalated SRT provided significantly higher PFS than lower-dose SRT within the group of 216 matched patient pairs (hazard ratio [HR] = 0.675; *p* = 0.0054). We found the same result after a repeated random combination of compatible match pairs, though the HRs and significance levels varied considerably. The same result was observed when the GS-matching criteria were changed from GS ≤ 7 vs. GS ≥ 8 to GS ≤ 6 vs. GS ≥ 7, which yielded 195 patient pairs (HR = 0.628, *p* = 0.0017, Figure 2b).

We then compared the different SRT doses in 387 patients (Figure 3a) that had received early SRT (PSA < 0.5 ng/mL). We found that the ≥7000 cGy dose improved the PFS (HR = 0.751) for patients with the significant risk factors identified in the multivariable Cox model (pT3–4, GS 8–10, R1), but the result was not statistically significant (*p* = 0.154). However, when we performed the analysis with propensity-matched patients (150 matched patient pairs, Figure 3b) we found a plausible improvement in PFS (HR = 0.719), with a trend towards significance (*p* = 0.059). 

Next, we evaluated factors that might influence OS with a multivariate analysis. We found that only age ≥64 years (HR = 2.16, *p* = 0.0051) and pT3–4 (HR = 1.97, *p* = 0.0133) could significantly adversely impact OS. Figure 4 shows the OS for the total patient cohort, stratified by (a) pT-stage and (b) SRT dose; only the pT stage impacted the OS. When we compared the pT2 and pT3–4 subgroups separately, we found that the different SRT doses did not significantly impact OS (Figure 4c,d). 

## 4. Discussion

After a failed RP, due to PSA persistence or relapse, escalating the SRT dose to >7000 cGy provided a significant advantage in PFS. We demonstrated this effect after adjusting for risk factors in a propensity-matched group of 432 patients. However, among patients with a pre-SRT PSA level < 0.5 ng/mL, we only observed a trend towards improved PFS with the escalated SRT dose. This lack of significance might have been due to the small number of patients in our subgroup. 

We found that dose escalations did not significantly improve OS. These results were consistent with several previous retrospective studies. However, a large systematic review of more than 10,000 patients found that, for every 100 cGy dose escalation, the freedom from biochemical recurrence improved by 2%. Therefore, those authors concluded that the applied SRT dose should be above 7000 cGy [21,22]. Nevertheless, there were substantial biases among the analyzed publications. For example, in 60 out of 71 studies, the median radiation dose was <7000 cGy; the patient characteristics were inhomogeneous, due to the inclusion of patients with positive lymph nodes; and a mean of 11% of patients (range: 0–90%) received ADT.

Early results are available from two randomized trials that investigated the effect of escalated radiation doses in SRT [17,18]. In both trials, biochemical progression was set as the primary endpoint.

In the SAKK 09/10 trial of the Swiss Cancer Foundation, 350 patients with biochemical progression after an RP were randomized to receive SRT, with either 6400 cGy (32 fractions) or 7000 cGy (35 fractions) delivered with an external beam and directed to the prostate bed [18]. The primary endpoint was freedom from biochemical progression (FFBP). The intent-to-treat analysis was performed for 344 patients. The authors defined PSA progression after surgery as two consecutive PSA rises, with a final PSA > 0.1 ng/mL, or three sequential PSA elevations. In addition, all patients had a post-operative PSA nadir of ≤0.4 ng/mL and a pre-randomization PSA of ≤2 ng/mL. After a median follow-up of 6.2 years, the dose-escalated SRT was not associated with improved outcome for any of the oncological endpoints, including the primary endpoint, the clinical PFS, the time to hormonal treatment, or the OS. The late toxicity analysis showed a significant increase in late grades 2 and 3 gastrointestinal toxicities (*p* = 0.009) in the dose-escalated group. Although no differences were found in terms of quality of life, the authors argued that patients with dose-escalated SRT were at risk of higher late GI toxicity without any oncologic benefit.

It remains unclear why the 6-Gy dose increase did not provide a measurable difference in PFS. According to the study by King et al., the dose escalation should have improved PFS by 2% per Gy of dose increment [21]. In that case, the SAKK trial should have observed a 12% improvement in PFS with the 6-Gy increase in SRT dose. One reason for the lack of a difference between groups may have been an insufficient follow-up. It is known that many local recurrences occur more than eight years after SRT. Therefore, it was possible that, after the median follow-up of six years, the 64-Gy dose may have postponed tumor progression, and the 70-Gy dose might have provided a local cure.

Differences in patient characteristics between our cohort and the cohort of the SAKK 09/10 trial and the Chinese trial are summarized in Table 3.

However, the well-considered definitions of inclusion criteria in the SAKK 09/10 trial might have led to a preselected patient cohort with a lower risk of progression compared to our cohort. Therefore, compared to our cohort, a higher proportion of patients in the SAKK-trial might not have needed an escalated radiation dose to prevent biochemical failure. Moreover, the large range of times from RP to SRT in our analysis could have resulted from the larger number of surgeons and/or variations in operative expertise. Both factors might also have introduced a selection bias in our study. For patients matched for pT2 vs. pT3–4, Gleason score ≤ 7 vs. ≥8, and surgical margins R0 vs. R1, escalating radiotherapy dose provides a near significant advantage in PFS (*p* = 0.059).

Another relevant factor influencing survival is PSA doubling time, as demonstrated in the systematic review by van den Broek [12]. In our patient cohort, data on the PSA doubling times were available only for a small subgroup of patients. Thus, we could not evaluate this risk factor.

In a Chinese phase III trial, 144 patients were randomly assigned to receive either 6600 cGy or 7200 cGy as an adjuvant for patients with high-risk factors (pT3–4, R1) or as salvage treatment for patients with a rising postoperative PSA of ≥0.2 ng/mL [17]. In the SAKK 09/10 trial, a higher overall dose did not provide an advantage for bPFS in the entire group, after a median follow-up of 48 months. However, in contrast to the SAKK-trial, in the Chinese trial, a subgroup analysis of patients at high risk (Gleason score 8–10) showed a significant improvement in the four-year bPFS with a 6-Gy dose escalation to 72 Gy (bPFS in 79.7% vs. 55.7% for 72-Gy vs. 66-Gy arms, respectively). That result agreed well with our findings, which showed five-year PFS rates of 52% and 65% for the <70 and ≥70 Gy groups, respectively (*p* = 0.0042). Moreover, in our investigation, we showed that the dose escalation was significantly beneficial, both for patients with Gleason scores of 8–10 and in patients with Gleason scores of 7 or higher. However, it should be noted that the different study designs of the Chinese trial and our study could have confounded the comparison; for example, the Chinese trial design included a shorter follow-up, the use of whole pelvic radiotherapy (88%), and the application of adjuvant radiotherapy (33% of patients). Moreover, due to substantial differences in cohorts and the possible introduction of a selection bias, a direct comparison between the Chinese trial and our cohort study is difficult. The Chinese trial included 48 patients with risk factors (pT3/4 or R1) that received adjuvant radiotherapy after an RP, without measurable PSA. Moreover, most of their patients (87.5%) received whole pelvic radiotherapy, and R1-resections were more frequent among patients in the high-dose cohort (64.4%) than among those in the low-dose cohort (47.9%, *p* = 0. 064). 

Although the patients in our cohort were highly homogeneous regarding the exclusions of pN+ disease and ADT applications, several studies have provided substantial evidence to show that some other pathological features can increase the risk of a selection bias. For instance, a previous retrospective analysis of 8770 patients showed that positive surgical margins (PSM) represented an independent predictor of biochemical failure [4]. In that study, among 579 patients that harbored PSMs, the likelihood of biochemical failure increased when the Gleason score was ≥4 at the margin, when the PSM was ≥3 mm long, and when the patient had multifocal positive margins. In the present study, these pathological details were not available for analysis, and this lack of data may have led to a substantial selection bias. Nevertheless, there are currently no data from randomized trials that focused on these aspects. Therefore, our results may be helpful when counseling patients in selecting an appropriate therapy.

Improving tumor control with higher radiation doses must be weighed against an increase in the occurrence of late toxicities. In this regard, our retrospective analysis may add further aspects to the current knowledge about variables that can influence dose escalations in SRT. 

Both randomized trials mentioned above completed patient recruitments in the pre-PSMA-PET-CT era; thus, they mainly performed conventional staging. The increasing implementation of advanced functional imaging, like PET-CT, might provide additional selection criteria for identifying patients with PSA progression after an RP and enhance the ability to detect risk factors that indicate the need for treatment intensification. For example, patients with local recurrence detected with PET-CT are likely to require a localized dose escalation in the future. Conversely, patients with no detection of macroscopic recurrent disease might be treated effectively with lower radiation doses. Thus, the available randomized data are of limited validity compared to data from patients staged with modern imaging.

In a recent retrospective study, 150 patients with local prostate cancer relapses detected in choline-PET-CT were treated with SRT. Radiation was delivered to the prostatic bed ± lymph nodes in 55% of patients, and the recurrent lesion received a local dose of 8000 cGy. Five- and seven-year relapse-free survival rates were 70% and 60%, respectively. Given the high radiation dose, grades 3 and 4 late toxicities were surprisingly low (2%) [23]. Those survival data were consistent with the survival rates we observed in the high-dose group in the present study, and both studies supported dose escalations.

The present study had several limitations. First, it was a retrospective study, with all the inherent limitations, compared to a randomized trial setting. Second, our data were not statistically suitable for correlation analyses with clinical outcome measures (e.g., metastasis-free survival), due to the limited number of patients with hematogenous metastases (*n* = 5, 1.7%). Third, for many patients (including patients with R1), irradiation was initiated when the PSA level was <0.2 ng/mL. This irradiation may indicate potential overtreatment, but only in a small number of patients that might have had benign, low-level, gradual PSA recurrences. Fourth, there is evidence that increasing the SRT radiation dose significantly increases the risk of late radiation GI toxicity. Yet, this difference was irrespective of treatment technique [18]. Due to the lack of toxicity data, our analysis could not assess the tradeoff between oncological benefits and radiation side effects. Fifth, the baseline PSA levels and GS values were significantly different between the two dose groups. Despite our attempt to minimize data distortions with propensity matching, there was a substantial risk of selection bias and systematic bias. Finally, there was also a risk of a treatment bias, due to the type of therapy; some patients received traditional 3-D conformal radiation, and others received modern IMRT.

This study also had some notable strengths. Despite the retrospective nature of our data, the matched pair analysis represented a decisive advantage. Moreover, we included a large patient cohort, with homogeneous patient characteristics, and homogeneous patient treatments. These factors reduced the biases inherent in retrospective data analyses. Furthermore, we excluded patients with positive lymph nodes and patients that received ADT or whole pelvic radiotherapy. In addition, we applied strict propensity-matching rules to the cohort.

## 5. Conclusions

This retrospective study demonstrated that a dose escalation above 7000 cGy had advantageous effects in patients with prostate cancer that underwent SRT for PSA persistence or relapse after an RP. Our findings contrasted with those of the SAKK-09/10 trial but were partly consistent with findings from a Chinese randomized phase III trial. Our additional risk-adjusted propensity analysis corroborated the finding that SRT had a beneficial impact on PFS. Differences between studies may be explained by different definitions regarding inclusion criteria, disease progression, or other patient-related characteristics. With longer follow-up times, the results of SRT dose escalations may become significant, as proposed by King et al. [21].

Future studies should focus on the evaluation of subgroups that may benefit from dose escalations, like high-risk patients, particularly when examined with advanced imaging methods, like PSMA-PET-CT or PET-MRI. With modern imaging, we may achieve a consensus definition of the ‘true’ relapse, which might change radiotherapy strategies substantially in the future. Moreover, studies should identify patient-related risk factors that might increase the risk of late toxicities. This knowledge will facilitate appropriate counseling for these patients by enabling a risk–benefit evaluation of dose-escalated SRT.

## Figures and Tables

**Figure 1 cancers-14-01320-f001:**
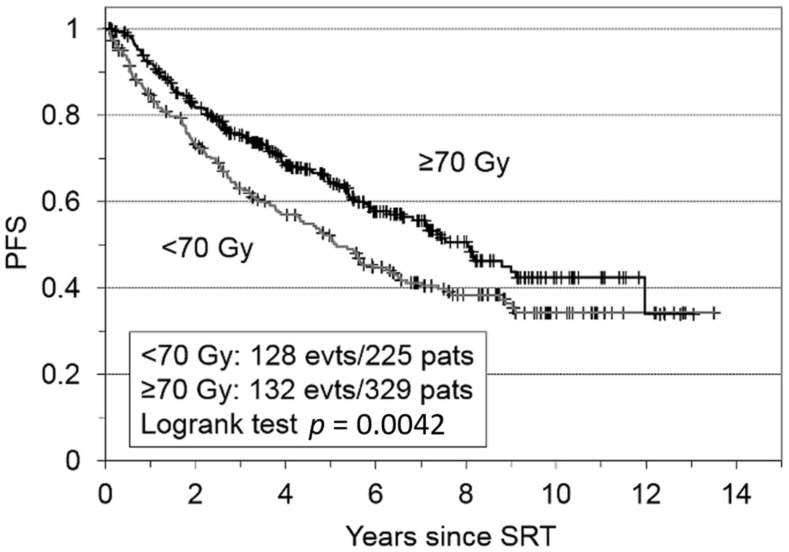
Kaplan-Meier plot shows progression-free survival (PFS) for the entire cohort of patients with failed prostatectomies (N = 554), after receiving salvage radiotherapy (SRT). Patients were grouped according to whether they received ≥70 Gy or <70 Gy.

**Figure 2 cancers-14-01320-f002:**
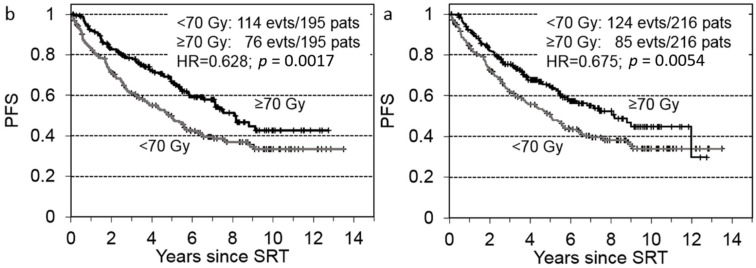
Progression-free survival (PFS) of propensity-matched patients with failed prostatectomies, after receiving salvage radiotherapy (SRT) delivered at ≥70 Gy or <70 Gy. Patients were propensity-matched 1:1 based on the following risk factors: pT2 vs. pT3–4, surgical margin status R0 vs. R1, and (**a**) Gleason score ≤ 7 vs. ≥8 (*n* = 216) or (**b**) Gleason score ≤ 6 vs. ≥7 (*n* = 195).

**Figure 3 cancers-14-01320-f003:**
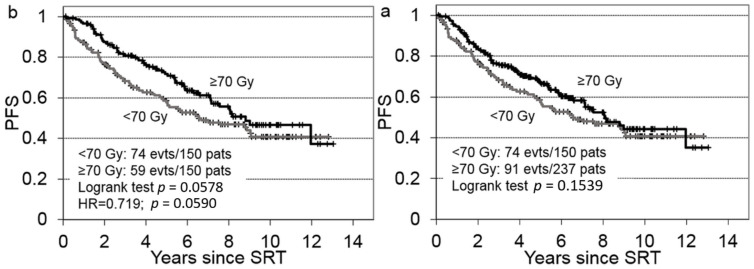
Progression-free survival (PFS) of patients with failed prostatectomies, after receiving early salvage radiotherapy (SRT) at a PSA < 0.5 ng/mL. SRT dosing groups (≥70 Gy or <70 Gy) were compared (**a**) before (*n* = 387 patients) and (**b**) after (*n* = 300 patients) propensity matching 1:1 for the following significant risk factors: pT2 vs. pT3–4, Gleason score ≤ 7 vs. ≥8, and surgical margins R0 vs. R1.

**Figure 4 cancers-14-01320-f004:**
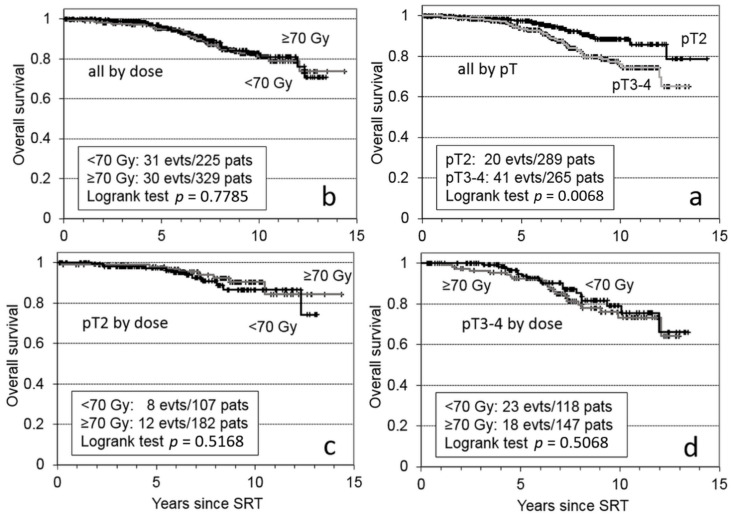
Kaplan-Meier plot shows overall survival of 554 patients with failed prostatectomies that received salvage radiotherapy (SRT). All 554 patients were stratified by (**a**) tumor stage (pT) and (**b**) SRT dose. Specific tumor-stage subgroups, (**c**) pT2 and (**d**) pT3–4, were stratified by SRT dose.

**Table 1 cancers-14-01320-t001:** Baseline characteristics of 554 patients with failed RP that received low-dose (<70 Gy) or high-dose (≥70 Gy) SRT.

Characteristic	<70 Gy (N = 225)	≥70 Gy (N = 329)	All (N = 554)
Age at RP, years; median (IQR)	63 (59–67)	64 (60–68)	64 (59–68)
Pre-RP PSA *, ng/mL;	10.00	8.87	9.40
median (IQR)	(7.00–15.16)	(5.98–14.4)	(6.28–14.7)
Tumor stage			
pT2	107 (48%)	182 (55%)	289 (52%)
pT3	114 (50%)	142 (43%)	256 (46%)
pT4	4 (2%)	5 (2%)	9 (2%)
Gleason score *			
GS ≤ 6	92 (41%)	96 (29%)	188 (34%)
GS = 7	87 (39%)	167 (51%)	254 (46%)
GS ≥ 8	46 (20%)	66 (20%)	112 (20%)
Surgical margins *			
R0	101 (45%)	201 (61%)	302 (55%)
R1	124 (55%)	128 (39%)	252 (45%)
Pre-SRT PSA, ng/mL;	0.294	0.290	0.292
median (IQR)	(0.140–0.690)	(0.180–0.516)	(0.160–0.568)

Values are the number of patients (%), unless indicated otherwise. RP = radical prostatectomy; PSA = Prostate Specific Antigen; IQR = inter-quartile range; GS = Gleason score; * significant difference between groups.

**Table 2 cancers-14-01320-t002:** Multivariable Cox regression analysis of potential risk factors for PFS in patients with failed RP that received SRT. Significant factors were used for propensity score matching.

Risk Factors	HR (95% CI)	*p*
Pre-RP PSA < 10 * vs. ≥10 ng/ml	1.14 (0.88–1.47)	0.3278
pT2 * vs. pT3–4	2.13 (1.62–2.79)	<0.0001
GS ≤ 7 * vs. GS 8–10	1.60 (1.20–2.14)	0.0015
Surgical margin R0 * vs. R1	0.68 (0.53–0.88)	0.0031
Pre-SRT PSA < 0.5 * vs. ≥0.5 ng/ml	1.56 (1.21–2.02)	0.0007

PFS = progression-free survival; SRT = salvage radiotherapy; RP = radical prostatectomy; PSA = Prostate Specific Antigen; HR = hazard ratio; GS = Gleason score; * State used for reference.

**Table 3 cancers-14-01320-t003:** Patient characteristics comparing the available randomized trails and our data.

	SAKK 09/10 [18]	Chinese Trial [17]	Own Data
Type of study	Open-label, multicenter Phase III trial	Randomized controlled Phase III trial	Retrospective cohort
Inclusion criteria	Biochemical failure after RP 3 PSA rises or 2 rises with last being 0.1 ng/mL Postoperative PSA-Nadir ≤ 0.4 ng/mL No ADT before or during SRT pT2a-3b No macroscopic relapse Nodal negative	Biochemical failure or PSA persistence after RP (ART/SRT = 48/96 Postoperative PSA-Nadir ≤ 0.4 ng/mL No ADT before or during SRT pT3–4 positive margin Nodal negative	Biochemical failure after RP PSA rise above 0.1 ng/mL No ADT before or during SRT pT3–4 positive margin Nodal negative
Treatment groups	6400 cGy vs. 7000 cGy Target volume: prostatic bed Technique: 3D CRT (44%), IMRT (57%) Assignment to treatment by randomization	6600 cGy vs. 7200 cGy Target volume: prostatic bed (RTOG-guideline) Technique: IG-IMRT/IG-VMAT High Risk patients: pelvic RT (88%)	<7000 cGy vs. ≥7000 cGy Target volume: prostatic bed +/− seminal vesicle bed (T3/4) Technique: 3D CRT (74.9%), IMRT (25.1%) Matched-Pair-Analysis
Primary endpoint	Freedom from biochemical Progression: Definition: PSA-increase ≥ 0.4 ng/mL beyond post-SRT-Nadir	Biochemical PFS: secondary therapy Definition: PSA-increase > 0.2 ng/mL beyond post-SRT-Nadir (x2), OS: death of any cause	PFS, secondary therapy Definition: PSA-increase > 0.2 ng/mL beyond post-SRT-Nadir OS: death of any cause
Secondary endpoints	Clinical PFS Time to hormonal therapy, OS Acute and late toxicity Quality of life	Acute and late toxicity Toxicity of hormonal treatment	n.s.
Number of patients	350 Conv. D.: 175 (170 ITT) Escal. D.: 175 (174 ITT)	144 Conv. D.: 71 Escal. D.: 73	554 low dose: 225 high dose: 329
Pre-SRT-PSA-level	0.3 ng/mL (0.03–1.61)	0.2 ng/ml	0.28 ng/mL (0.04–8.87)
Follow-Up	6.2 years (IQR 5.5–7.2)	48.5 months (14–79 months)	6.8 years (IQR 4.2–9.2)
Time RP–SRT	6400 cGy: 25.9 mo. (14.0–42.3) 7000 cGy: 30.3 mo. (15.8–50.8)	8 mo.	23 mo. (1.7–176)
Results	Reported: 6-year-results 6400 cGy: 62.3% (95% CI: 54.2–69.4) 7000 cGy: 61.3% (95% CI: 53.4–68.3) bPFS: *p* = 0.44 Hazard-ratio: 1.14 (95% CI: 0.82–1.6)	Reported: 4-year-results 66 Gy: 75.9%; 95% CI, 71.6–79.6% 72 Gy: 82.6%; 95% CI, 78.8–85.7% bPFS: *p* = 0.299	Reported: 5-year-results Conventional dose: 52% Escalated dose: 65% PFS: *p* = 0.0042 Multivariate Analysis: significant improvement favoring: lower pT stage lower Gleason sum positive resection status lower-pre-SRT PSA level

Abbreviations: RP—radical prostatectomy; PSA—prostatic specific antigen; ADT—androgen deprivation therapy; SRT—salvage radiotherapy; cGy—centiGray; RTOG—Radiotherapy Oncology Group; 3D CRT-three—dimensional conformal radiotherapy; IG—image guided; IMRT—intensity modulated radiotherapy; VMAT; RT—radiotherapy; PFS—progression free survival; OS—overall survival; Conv. D.—conventional dose; Escal. D.—escalated dose; ITT—intention-to-treat; IQR—inter quartile range; bPFS—biochemical progression free survival; mo.—months; CI—confidence interval.

## Data Availability

The datasets generated during and/or analyzed during the current study are available from the corresponding author on reasonable request.

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
