# Peer review of "Impact of Dose Escalation on the Efficacy of Salvage Radiotherapy for Recurrent Prostate Cancer—A Risk-Adjusted, Matched-Pair Analysis"

_cancers, 2022, doi:10.3390/cancers14051320_

Round 1

Reviewer 1 Report

I have read with great interest this paper concerning dose escalation for salvage radiotherapy in recurrent prostate cancer. Data are properly collected and clearly presented. Only some issues:

  • Table 2 shows a HR of 0.68 in favour of R0 surgical margin as result of the multivariable Analysis. Please comment on this.
  • Line 294: A near significant advantage in PFS with a p-value of 0.059 is described for patients with "high-risk factors" in your cohort. This refers to the analysis you did after propensity matching including patients with pT2, Gleason score ≤ 7 and R0-resection as well. Please clarify.

Author Response

Dear Reviewer 1

Thank you for reviewing our manuscript.

Point 1: Table 2 shows a HR of 0.68 in favour of R0 surgical margin as result of the multivariable Analysis. Please comment on this.

Thank you for making this important point. Indeed, progression free survival is improved for patients with R1 resection. These patients have a high risk of bearing tumor cells in the prostatic bed and thus radiotherapy to this area results in improved PFS. In contrast, patients with R0-resection and rising PSA values may harbor tumor cells outside the prostatic bed more often than their R1-counterparts.

Point 2: A near significant advantage in PFS with a p-value of 0.059 is described for patients with "high-risk factors" in your cohort. This refers to the analysis you did after propensity matching including patients with pT2, Gleason score ≤ 7 and R0-resection as well. Please clarify.

Thank you very much for addressing this point, which we did not present clearly. We have changed the according sentence: "For patients matched for pT2 vs. pT3-4, Gleason score ≤7 vs. ≥8, and surgical margins R0 vs. R1, escalating radiotherapy dose provides a near significant advantage in PFS (p = 0.059)."

Reviewer 2 Report

Author Bohmer et al., demonstrated the impact of salvage radiotherapy for recurrent prostate cancer in 554 patient cohorts, where after radical surgery the persistent increase in PSA levels required salvage radiotherapy. The author showed that increasing the dose during radiotherapy could reduce the risk of rising PSA or relapse. “Higher dose of radiotherapy significantly reduces the tumor spread” but still that didn’t translate into better survival.  The previous finding with a high escalating dose did not observe any statistical significance particularly with overall survival (OS) and which is in agreement with the current study. Still, the insignificance difference with overall survival in patients undergoing high escalating doses is critically important, considering OS was the primary endpoint.  Most importantly, it is extremely difficult to safely deliver high doses of radiation to the prostate gland without excessive toxic effects, and it is very important to develop treatment plans for each patient on a case-by-case basis.

Major concern:

  1. Research findings from the Washington University of Medicine (published in JAMA oncology) 2015 showed that higher doses of radiation don’t improve survival in prostate cancer patients compared with the standard radiation treatment.
  2. How about the side effect in the escalating dose group such as urinary irritation or rectal bleeding? Line 363-364 author mentioned that analysis could not assess the tradeoff between oncological benefits and radiation side effects.
  3. When the author analyze the data how did they minimize the treatment biasness, as some patients were treated with traditional 3-D conformal radiation, and others received modern IMRT.
  4. Do the patients in the standard or low dose groups undergo further therapies to control tumor growth?
  5. It will be more supportive to add the data for risk of severe morbidity (physical and clinical factors), Gastrointestinal (GI), and Genitourinary (GU) toxic effects for different grades.

Author Response

Dear Reviewer 2

Thank you for your valuable comments.

Point 1: Research findings from the Washington University of Medicine (published in JAMA oncology) 2015 showed that higher doses of radiation don’t improve survival in prostate cancer patients compared with the standard radiation treatment.

Unfortunately, we could not find the 2015 article in Jama Oncology that you mentioned regarding salvage RT dose escalation. Anyway, you raise an important point regarding the relevance of survival as an outcome parameter. In the two randomized trials (Ghadjar et al. and Qi et al.) the authors found no significant improvement of dose escalation regarding all oncologic outcome parameters in the respective patient cohorts. Yet, there was a significant difference for bPFS in the Gleason 8-10 subgroup in the Chinese trial.

We believe that there are patient subgroups that benefit from dose escalation, and we hope to contribute to defining stratification and risk parameters with our analysis.

Point 2: How about the side effect in the escalating dose group such as urinary irritation or rectal bleeding? Line 363-364 author mentioned that analysis could not assess the tradeoff between oncological benefits and radiation side effects.

Thank you for this crucial point. We are aware that this is a shortcoming of our study. Lacking data regarding toxicity in our patients do not allow any statement regarding this tradeoff. Fortunately, both randomized trials have analyzed toxicity in detail. In summary, only the SAKK-Trial (Ghadjar et al.) found a significant increase in late GI toxicity, but this was irrespective of the treatment technique.

We have changed the sentence line 363-364 to: “Fourth, there is evidence that increasing the SRT radiation dose significantly increases the risk of late radiation GI toxicity. Yet, this difference was irrespective of treatment technique.”

Point 3: When the author analyze the data how did they minimize the treatment biasness, as some patients were treated with traditional 3-D conformal radiation, and others received modern IMRT.

Thank you for this important statement. We agree that the type of radiotherapy (3-D conformal vs. IMRT) is relevant regarding GI and Gu toxicity. Regarding the primary outcome measures of our study (oncological outcome and defining factors that influence oncologic outcome with respect to dose escalation), the number of patients receiving 3-D RT and IMRT are less relevant. We have added the reference to treatment technique in the sentence line 363-384 as follows: “Fourth, there is evidence that increasing the SRT radiation dose significantly increases the risk of late radiation GI toxicity. Yet, this difference was irrespective of treatment technique.”

Point 4: Do the patients in the standard or low dose groups undergo further therapies to control tumor growth?

Thank you for addressing this issue.

Patients in the low dose arm received radiotherapy only, as did their counterparts in the high dose group. Patients with any additional treatment, such as hormonal therapy or chemotherapy were excluded from analysis.

In general, data on subsequent therapies are of crucial relevance when assessing clinical treatment results. In the limitations section we have pointed out that missing data do not allow for an estimation of clinical outcome measures.

Point 5: It will be more supportive to add the data for risk of severe morbidity (physical and clinical factors), Gastrointestinal (GI), and Genitourinary (GU) toxic effects for different grades.

Thank you for addressing this point which relates to your point 3 above. We agree with you that the manuscript would gain substantially if data on late toxicities were available. Due to lacking data, we only can refer to the data of randomized trials, as we have elaborated in point 3 above. 

Reviewer 3 Report

This paper deals with a retrospective analysis of the effects that higher and lower doses than 70 Gy have for prostate cancer salvage radiation therapy, regarding several prognostic factors for patient classification. The language is adequate and the subject under investigation is relevant with the journal’s thematology. Despite having several limitations, this study would be of clinical interest to the readership since the statistical analysis was performed for homogeneous subgroups of patients and several variables were taken into account. 

I’d like the authors to address the following.

The patients included in this study were treated between the years of 1997 and 2017. The radiation treatment techniques have evolved a lot technologically during that period, resulting in a high dependance of the toxicity or the progression free survival, on the therapy that the patients received, namely 3DCRT, IMRT or VMAT. In line 87 the authors mention that more than half of all the patients received therapy with IMRT technique and in Table 3 it is mentioned that the treatment techniques were 3DCRT or IMRT. Both are modern radiotherapy techniques, however the low dose bath as well as the Organs at Risk irradiation can differ a lot for these two. It would be better if the patient groups were further categorized according to the technique for the results to be even more beneficial. The authors should mention in the manuscript the ratio of the patients that were treated with each technique, so as to establish the balance between the groups. A discussion and some minor comments on this would greatly help the manuscript.

Author Response

Dear Reviewer 3

Thank you for your constructive review.

Your points of critique: The patients included in this study were treated between the years of 1997 and 2017. The radiation treatment techniques have evolved a lot technologically during that period, resulting in a high dependance of the toxicity or the progression free survival, on the therapy that the patients received, namely 3DCRT, IMRT or VMAT. In line 87 the authors mention that more than half of all the patients received therapy with IMRT technique and in Table 3 it is mentioned that the treatment techniques were 3DCRT or IMRT. Both are modern radiotherapy techniques, however the low dose bath as well as the Organs at Risk irradiation can differ a lot for these two. It would be better if the patient groups were further categorized according to the technique for the results to be even more beneficial. The authors should mention in the manuscript the ratio of the patients that were treated with each technique, so as to establish the balance between the groups. A discussion and some minor comments on this would greatly help the manuscript.

Thank you very much for bringing up the important issues of radiation technique and toxicity. The team has been discussing this extensively. We agree that a separate analyses of 3-D and IMRT/VMAT treated patients would allow for a more detailed insight into the relevance of radiotherapy technique on outcome measures.

Unfortunately, data regarding acute and foremost late toxicity are not available. Moreover, the matched pair approach regarding dose escalation would not allow a toxicity analysis without accepting the risk of severe selection and reporting bias. Secondly, reducing the number of patients in each matched group by separating 3D conformal RT patients from those who received IMRT or VMAT would results in subgroups too small to detect a significant difference in our primary outcome measure, oncological outcome.

Your next point regarding treatment technique is of major importance. We agree that the type of radiotherapy (3-D conformal vs. IMRT) may be relevant regarding GI and GU toxicity and oncological outcome. Regarding the primary outcome measures of our study (oncological outcome and defining factors that influence oncologic outcome with respect to dose escalation), the number of patients receiving 3-D RT and IMRT are less relevant. We have added the reference to treatment technique in the sentence line 363-384 as follows: “Fourth, there is evidence that increasing the SRT radiation dose significantly increases the risk of late radiation GI toxicity. Yet, this difference was irrespective of treatment technique (Ghadjar et al.).

We also added the reference to table 3, where we added the numbers of patients who received 3-D conformal or IMRT.